# Assistive Technology during the COVID-19 Global Pandemic: The Roles of Government and Civil Society in Fulfilling the Social Contract

**DOI:** 10.3390/ijerph182212031

**Published:** 2021-11-16

**Authors:** Daniel Mont, Natasha Layton, Louise Puli, Shivani Gupta, Abner Manlapaz, Kylie Shae, Emma Tebbutt, Irene Calvo, Mahpekay Sidiqy, Kudakwashe Dube, Ulamila Kacilala

**Affiliations:** 1Center for Inclusive Policy, Washington, DC 20005, USA; Shivani.gupta@cbm.org (S.G.); abner.manlapaz@inclusive-policy.org (A.M.); 2Rehabilitation, Ageing and Independent Living (RAIL) Research Centre, Monash University, Clayton, Melbourne 3800, Australia; natasha.layton@monash.edu; 3Access to Assistive Technology Team, World Health Organization, 1211 Geneva, Switzerland; pulil@who.int (L.P.); shaek@who.int (K.S.); tebbutte@who.int (E.T.); calvoi@who.int (I.C.); 4Kabul Orthopaedic Organization, Kabul 1001, Afghanistan; mahpekay.sidiqy@yahoo.com; 5Africa Disability Alliance, Pretoria 0077, South Africa; akdube@samaita.co.za; 6Pacific Disability Forum, GPO, Suva Box 18458, Fiji; ulamilakacilala@gmail.com

**Keywords:** assistive technology, policy, disability, assistive products, service provision, health systems, COVID-19

## Abstract

The COVID-19 pandemic imposed significant challenges to users of assistive technology (AT). Three key issues emerged from a series of structured qualitative interviews with 35 AT users in six low- and middle-income countries. These were (1) access to information about COVID-19 and available supports and policies, (2) insufficiency of the government response to meet the needs of AT users, and (3) the response of civil society which partially offset the gap in government response. AT users noted the need for better communication, improved planning for the delivery and maintenance of AT during times of crisis, and higher-quality standards to ensure the availability of appropriate technology.

## 1. Introduction

A pandemic is a classic instance necessitating a public health response. A key component of that response is getting information out to the public [1], and when that pandemic impinges on economic activity and supply chains, the response must also consider services to people to meet increased needs. Those needs can be directly caused by the disease but also can result from the responses needed to meet the crisis, such as lockdowns and the associated economic crisis [2,3,4]. This is particularly important for people who rely on assistive technology (AT) and support services [5].

AT includes the systems and services by which people obtain assistive products. Assistive products, such as wheelchairs, prostheses, and glasses, improve people’s ability to function and enable them to live independent lives and participate in desired activities. There are many types of assistive products, so, to assist in prioritization, the World Health Organization has published a list of the fifty most needed assistive products [6]. WHO also provides guidance on the necessary steps in assistive product provision, including selecting the appropriate products through screening and assessment, fitting or providing the product, training the person on how to use the product, and follow up for review and maintenance.

The disruption of the systems that provide and maintain their assistive products has the potential for serious consequences. The current corona virus pandemic (COVID-19) is going to be with us for some time to come, and therefore the lessons learned from interviews with AT users from low- and middle-income countries and reported in this paper can inform ongoing responses to the current situation, as well as preparation for future pandemics and other emergencies.

The public health response to COVID-19 was of particular importance to people who use AT for a number of reasons. First, many AT users have pre-existing health concerns that may render them more susceptible to COVID-19 [7,8,9]. In the UK, for example, between January and November 2020, the risk of death due to COVID-19 was 3.1 times greater for people identified as having a higher level of disability (3.1 times greater for men and 3.5 times greater for women) in comparison to people without a disability. Even for people classified as “less-disabled”, death rates were about double those for people without disabilities [10]. Second, people who use AT are often from vulnerable groups within the population, such as those living with disabilities, chronic illness, or the impacts of ageing, who tend to have less access to resources, especially during difficult economic times [11]. Third, they rely on AT and related support services, for which new access barriers arose during the public health response [12].

This study hears directly from AT users in low- and middle- income countries about their experiences with AT services and related support services during COVID-19. It is part of a series that draws from interviews and survey responses from AT users and providers to better understand the overall impact of COVID-19 [2,13]. This article focuses on the themes that emerged from the interviews about the appropriateness of government responses and what other supports and capabilities enabled people to survive and flourish. Overall, regarding both of these things, interviewees reported that the government response had to be supplemented significantly not only by family and friends, but also through coordinated efforts by the local community, including organizations of persons with disabilities, to meet their specific needs.

## 2. Materials and Methods

Qualitative methods were employed to uncover and describe the impacts of the pandemic on AT users. This method uses purposive sampling and inductive analyses to uncover the dynamics of peoples’ lived experience and provide insight into the processes affecting peoples’ lives [14,15].

The first step in this process was a rapid literature review to identify themes related to the impact of COVID-19 on AT, ageing, and disability. These were used to develop an interview guide used to collect qualitative information from AT users about their experience during the pandemic (Appendix A).

A global research team was assembled including experts in providing or using AT, disability, and policy research. This team comprised researchers from two academic institutions, two international non-governmental organizations, from WHO’s assistive technology team, and researchers based in each of the six WHO regions.

The regional researchers were selected based on the fact that they were users of AT and support services themselves, were very familiar with their country culture and context, and had strong connections with the disability community so they could locate a diverse set of interviewees. Such firsthand knowledge was considered to improve the comfort level of the interviewees and the quality of the interviews. Interviewees were chosen as part of a purposive sample to capture a range of experiences, based on age, gender, and type of AT used.

The semi-structured interview guide (Appendix A) asked about basic characteristics (age, rural/urban, household structure, nature of their functional difficulties) and the AT and support services used by interviewees pre-COVID-19. Categories of difficulty experienced by the interviews align with those in the Washington Group short set of questions on disability with the addition of psychosocial. The guide then asked about the COVID-19 experience, including the sources and nature of information received about COVID-19, the problems with accessing AT and support services, the perceived cause of those problems, and the responses by family and friends, the community, and government to ameliorate them. The guide also asked about the personal strengths, previous experiences, and personal resources that helped the interviewees deal with the problems they faced and ended with questions to identify their biggest ongoing needs and to raise issues not addressed by the field guide. Thematic analysis uncovered the themes and sub-themes shown in Table 1.

Upon receipt of interview reports, an initial set of codes was developed based on a subset of interviews by one researcher, who then conferred with another researcher who independently reviewed the same set of initial interviews. The codes (shown below) were then expanded and finalized based on the full set of interviews. An initial report of the interview findings was then shared with the researchers who carried out the interviews to validate that the findings were consistent with their experiences interviewing the respondents.

Data were handled and retained in accordance with Monash University ethical requirements. All interviews were conducted between 1 December 2020 and March 2021. The data were analyzed using the qualitative data analysis software NVivo (QSR International, London, UK).

One limitation was that the sample was mainly drawn through contacts with organization of persons with disabilities and community organization networks, although regional researchers were encouraged to reach out beyond their usual networks. In addition, the fact that the interviews needed to be conducted remotely also affected sample selection.

## 3. Results

### 3.1. Results—Quantitative

Respondents (*n* = 35) to the interview questions were evenly split between men (18) and women (17), with six interviews from five countries (Afghanistan, Fiji, India, Philippines, South Africa) and five interviews from Peru. Twenty-three were from urban areas and 12 from rural areas. Breakdowns by age and type of functioning difficulties experienced are shown in Table 2. The type of functional difficulty is based on interviewee self-reports when asked about their disability and/or AT needs. Since some people reported experiencing difficulties (or limitations) in more than one area, the sum by type of difficulty exceeds 35.

### 3.2. Results—Qualitative

Collectively the 35 interviewees provided a wide range of information on the role of AT in their lives and how the COVID-19 pandemic affected them. Many of those themes also emerged from the AT user/family and AT service provider survey components of this project explained in the methods section of this paper. These are elaborated in sister articles. The results presented here focus on two main themes explored in depth during the interviews: access to information and the government and civil society response to the pandemic.

#### Information

The AT users interviewed reported that their first sources of information were from the government, but as explained below these were insufficient. Therefore, they relied primarily on social media, until in some places local networks developed to provide good information. While government sources were respected, they were harder to access. In contrast, social media was easier to access than government sources; however, the reliability was sometimes suspect. Where local networks developed, they were more trusted and more tailored due to a better understanding of people’s needs.

In the first few months of the pandemic, AT users interviewed reported that there were a lot of rumors. Correct information was not reaching their communities. At first the main sources of information were from the government through local media—radio and television.

*I first received information about the COVID-19 outbreak from the media then the government and health agencies,* (I01, Fiji).

*The main source of information is Radio-BBC World Service and Indian National and regional radio stations. It has been so from the time I remember. I will be 74 in May 2021,* (I02, India).

This information was often not accessible for people with hearing or visual difficulties. For those with hearing difficulty, accessible communication formats such as sign language or captioning was generally not available. People with visual difficulties also highlighted the importance to them of having accurate information given the necessity for using their hands, along with white canes to navigate environments, increasing contact with surfaces and, therefore, susceptibility of contracting COVID-19 [16]. Despite this risk, much information was transmitted through the use of videos without audio descriptions adequate for people with vision difficulties.

*The media released the information and protection guides on how a person her/himself [can protect themselves] from COVID-19 like don’t participate in public event or don’t hold any event and wash hands with cleansers regularly, use mask, (…) to recognize COVID-19 and to be aware to save ourselves and others, but unfortunately the government and SCO [state coordinating officer] didn’t consider people with vision disabilities. All the information was for others and not for us. We can’t see the video to know the how to wash the hands and other essential information and we don’t have access to protective equipment and sanitizers which we seriously needed because we use the assistive device [white cane] and it is made of metal and my hand touches the stick, so need to wash it every time I use it,* (I21, Afghanistan).

In the end, most people relied mainly on social media for their information because of the perceived insufficiency of official communications. Social media, primarily through their phones, was used to connect to news feeds but also to connect with family and friends.

*Nowadays social media is very common so most information about COVID-19 I got from there,* (I4, India).

*My past experience in communicating, the use of technology and my knowledge of current events helps me deal with the pandemic. It helps me deal with the new way of life like working from home,* (I05, Philippines).

Given that social media was mentioned by six (17%) of the respondents as the main way that information was obtained by themselves and others, and that it was mainly accessed through phones, adequate access to smartphones and internet access was critical. However, access to this method of accessing communication depended on people’s income, the availability of a good internet connection, and their technical knowhow. So, while it worked well for people with these things in place, others were excluded.

*Most of us were dependent only on the phone. We started phone counselling to reach out to people. But not many people have smart phones with adequate internet on it, including our staff. Most people have a very basic plan, with an old phone. Also, they could not buy a data card because it was not a priority for them. What they needed was food. So, they could not reach out to anybody,* (I6, India)

*Assistive communication devices such as live transcription on my cell phone, but these are not always accessible if I am in a range where there is no WIFI signal. The App is free, I downloaded it online but one must always have WIFI connectivity to use it,* (I7, South Africa).

However, while people relied on social media, they worried at times about the veracity of the information they were receiving. Steady access to a reliable source of information from the government was not easily obtained.

*There’s a lot of information in the Facebook which makes her doubt what she is reading,* (I08, Philippines).

Another complicating factor was that the need for social distancing deprived some people of the interpreters they needed to obtain information. As one deaf-blind man reported.

*I have not been able to receive much information since I depend on my guide interpreter (…) During the pandemic caused by COVID, particularly during the lockdown, I was not able to have access to information*, (I09, Peru).

Ultimately, many AT users reported that they received the information they needed from their local disability networks and their immediate family and friends, in addition to government sources. This was easier for people whose families were more educated or who were living in areas, sometimes abroad, where information was more easily obtained. People living in urban areas also benefitted from more developed disability networks.

*All the information that I received about COVID-19 was from the disability network and not from the health centre (…) My father is a dentist. He attended a lot of seminars on COVID-19 and he was able to share all that information with us and made us more careful. Because we are careful, we are managing very well,* (I10, India).

*Training was provided to members of the Psychiatric Survivors’ Association about COVID-19 and safety precaution measures,* (I11, Fiji).

*[Locals CSOs]… provides information. NGO conducted awareness-raising on how to protect oneself from COVID-19. Facebook, radio, is also an important source of information,* (I12, Philippines)

*For example, what is social distancing? The typical videos sent out were largely in English, also people do not have sense of what six feet means in physical space. So, we made some videos to simplify and make it visually clear. We also had a voice over for people who cannot see. We used some ABT (Arts based therapy) techniques to show what is six feet. It was in local language voice over. We also ensured that it was compatible to be received by most phones. What we found was while the government and other resources were very good, they were too technical and not very relevant to our communities [they] did not address the cultural rumors. So, we started our own campaign to bring information about Corona and the safety measures to our communities and people who are registered with our program. We did that with our own resources (…) Applications like WhatsApp were very useful for us in the mental health sector to keep in contact with our peers and support systems. It enables us not to be visible in front of people* (I06, India).

### 3.3. Government Assistance

Unfortunately, interviewees found that, aside from some cash benefits and minor subsidies for their assistive products, government assistance was not sufficient.

*The government or an NGO didn’t help me, I request help from the government, but they ignored me, they even insulted me,* (I13, Afghanistan).

*We didn’t receive any help from government, just the organization where I am working with (…) and a small donation from a civil society organization. Because I lost my job and I was faced with financial problems beside being infected by COVID-19,* (I14, Afghanistan).

*I did not receive any economical compensation of any NGO or government since I was working from home and the government aids were only for unemployed persons. They [government aid] were not successful since neither the government nor the NGOs offered eyecare.* (I15, Peru).

This was particularly problematic for AT users because of the extra costs associated with their AT use, which was often not covered. They reported that the government did not have programs covering the costs that they were incurring related to their AT, for example, transportation and maintenance and repair of devices. These costs, of course, exist regardless of COVID-19 but were exacerbated by conditions during the pandemic, for example, increased difficulties in using taxis or public transportation.

*I am also worried on the additional cost that I have to pay to get the prosthesis because [Government provider] does not cover all the cost of the assistive device if it exceeds the allowable program package costs* (I12, Philippines).

*My orthosis was breaking, and I was not able to walk with it. I needed the government and NGOs. I didn’t have a possibility at least to repair my orthosis in some private clinics that was open at that moment. Due to the poor economic situation we have, I was not able to do it by myself in private orthopedic centers. My husband tried to repair my orthosis. He had taken some metal and he made a knee joint from metal and he tightened it with wire on my orthosis. It worked for two weeks but again it broke and after that I moved like a child, crawling this way I did all chores and my husband help me for seven months I moved like this* (I16, Afghanistan).

The uncertainty of when their AT could be followed up, maintained, or replaced also added to their mental stress in addition to affecting their ability to physically function.

### 3.4. Accessing Benefits and Services

Even when more empowered people with health systems knowledge and social connections attempted to help people access benefits, they often experienced significant barriers. Only people with connections to organizations of persons with disabilities (OPD) or others with the capacity to effectively put pressure on the government got what they needed, worsening inequality. While this affects all people, it is particularly acute for AT users who may be more isolated and, on average, more in need of services.

*We could contact the government helpline and we helped others to contact the helpline and also to get rations and the 1000 rs benefit from the government. The government was helpful but there were issues in accessing these benefits…from across the state most people felt that they were unable to connect. Those who could not get through posted on our WhatsApp group and we had to bring it to the notice of the disability commissioner who asked the helpline staff to call back the persons with disabilities. Even for getting the rations and the INR 1000/- they told us on the phone to come to the nearby corporation office and take these. I said no because of my disability I cannot step out of the house. After negotiating with them for more than a week they got it at home. Many people had this issue. Many people went to the corporation office to get it. The disability commissioners order however, was to provide it on the doorstep. Here again we have to bring this to the notice of the commissioner who solved the problem individually. So, the government order says something and there are great things on paper but it’s not being implemented effectively on ground* (I10, India).

As a result, interviewees reported that it was civil society, including non-governmental organizations, community groups, faith based-organizations, volunteers, and families, that stepped in to fill the gap. This works in areas with developed disability and similar networks but not for the most excluded people or people living in remote areas. This is an additional way in which COVID-19 has exacerbated inequality and exposed cracks in society, while at the same time providing essential services to some.

*Wheelchairs were provided by the organization free of charge by the Fijians in the UK, counterparts in New Zealand and counterparts in Australia through [wheelchair donor]. They provided support to their family as they are like an extended family* (I17, Fiji).

*For Deaf people the masks have been the biggest barrier. Deaf entrepreneurs became innovative and came up with masks with a see-through portion so people with hearing loss can lip read but these initiatives have not received much support from the government. The expectation was that these types of masks would be regulated into the procurement policies of government departments so that all health care facilities and other government service centres could utilize these types of masks to ensure that people with hearing loss are able to continue receiving services they need but this has not happened, so they continue to be barred from accessing information and receiving full public services* (I7, South Africa).

*She goes to church, conducts organizing of persons with disabilities, meeting government people. Persons with disabilities in their municipality received assistance from [several civil society and faith based organizations]* (I18, Philippines).

The most extensive examples of this were reported in India, where one interviewee who works with an organization for people with disabilities reported that even though they had never done relief work before

*(…) we had to provide food to families. We also created a cash transfer system so that people can buy what they want. They can also buy a gas cylinder for which people didn’t have the money* (I6, India).

He went on to say that for people with psychosocial disabilities it was important that they provide “a message of hope”. Because of social distancing and lockdown procedures they had to be creative. For example,

*(…) one activity called ‘chalta bolta activity’ where one person walks around the slum lanes, stops with one person at a time and have a brief exchange telling them they are from Bapu Trust and bring a message of hope and share news about mental health and wellbeing during corona times and ask some questions. Community people felt supported and happy to get a message of hope. We were recognised as an essential service by the government, so our field staff was allowed to go in the community* (I6, India).

Another respondent working in India reported that she was in touch with an NGO that

*(…) had a very good initiative to help persons with disabilities and older persons to get daily essentials at their doorstep. They had a large number of volunteers and a couple of times even we ordered. They did a very good job. I had to just WhatsApp a list of items to them and sometimes within two hours they used to deliver. I was able to refer some other families who were unable to go out due to disability or age to them. For them Project Mumbai was God sent (…) In between I had to increase me medicine dosage because of all these symptoms and vision was also fluctuating* (I19, India).

They were also able to set up online therapy sessions and set up other services such as

*A volunteer for her to buy provisions for her. What I see is people’s intrinsic needs are understood only by the community-based organizations who work closely with any specific community (…) we just share information with each other which gave a lot of strengt’* (I19, India).

The same interviewee also cited the flexibility of the private sector in helping during the crisis. While these accommodations were no doubt extremely useful, they were probably applied unevenly and based on levels of income and social capital that are not available to everyone.

*When I told the dealer that I wanted this extra belt on the footrest of my new electric wheelchair and spoke to the engineer, they never said no and were ready to serve me and never said that because of the lockdown we cannot come. Fortunately, [they] had a pass to come out during the lockdown because they are dealing with health care. When they entered my apartment building, I made sure that the wheelchair was kept on the floor lobby in advance. So they did not come into my house. So, they did whatever they had to outside my house. I wore a face shield and a mask and even the engineer did the same* (I10, India).

While the respondents from India reported the most developed systems of civil society support, such support was forthcoming in other countries, too.

*We didn’t receive any help from government, just the organization where I am working with, had help me with small donation and a small donation from a civil society organization. Because I lost my job and I had faced with financial problem beside that we infected by COVID-19* (I14, Afghanistan).

*Private donors assisted with food and R3.5 million (US$240 740) for person with disabilities in Gauteng province* (I20, South Africa).

### 3.5. Policy Reforms

The majority of interviews pointed to the need for policy reforms that could ensure a more timely and appropriate response to AT users during times of crisis. Moreover, those reforms would also contribute to service delivery in general. These responses fell into three key areas:Removing communication barriers,Improving planning, andDeveloping standards of quality and accessibility.

#### 3.5.1. Removing Communication Barriers

Interviewees believed that all information should be made accessible to people with all types of communication needs. This includes sign language but also voice over descriptions of videos for blind people and easy to read messaging for people with intellectual disabilities. Access to smart phones equipped with apps that enable communication in their desired formats. Because much communication is not accessible for people with visual impairments, having a phone that is capable of overcoming some communication barriers (e.g., a screen reader or texting capabilities) is a necessity, not a luxury.

*Barriers of communication should be removed, and they should be sign language interpreter is available for sample during the COVID-19 time there were many deaf people who did not even know what is happening because the families do not know sign language, so they were unable to communicate with anyone and it did not understand fully what had happened and why they were living in the lock down* (I4, India).

#### 3.5.2. Improved Planning

People who use AT often require assistive products for participation in everyday life, be it employment, education, or the community. AT services are essential to their health and wellbeing. As such, improved government knowledge of AT users’ specific needs and locations was suggested by interviewees as something that may support a more efficient and targeted response.

*There should be a database of all persons with disabilities and the support means available with the block level and panchayat level government especially because [we are] susceptible to a lot of disasters. For instance, sometime back that was a flood and persons with disabilities lost their wheelchairs and other assistive devices in that. For nearly six months they did not get any relief, they were being carried in laps and their dignity was not upheld. Secondly at the community level there should be something like a disaster management group, and they should be given proper training for finding persons with disabilities, taking them to the hospital, shifting them to the places of rescue. We also do not have proper planning of distribution of these assistive devices sometimes hearing aid is developing more complications for the hard of hearing person instead of helping them to listen (…) Therefore, proper system and assessment, technology, materials used must be [prepared]* (I4, India).

*A better health system is needed in the future to have an emergency service provision plan like (mobile team) if there is a lockdown or movement restriction this team needs to go to the region to help with vulnerable people* (I14, Afghanistan).

The ability to respond quickly is especially important for people in remote areas or living furthest from established AT service hubs. The impact of disruption to global supply chains caused by an event such as the current pandemic is heightened in countries that have limited local production and/or pre-positioned stock of assistive products and spare parts. Distance and supply issues lead to longer waits for assistance, which is not merely an inconvenience, but can seriously impact a person’s ability to function and lead to a worsening of the underlying impairment.

*The biggest need for the organization is getting the equipment on time to Fiji. We don’t produce the equipment. The challenge [is that] shipments with that product will be imported from abroad and also the collection of this equipment from abroad and the arrival of product into the port has become expensive for the organization and the countries like Fiji that are at the mercy of donors* (I22, Fiji).

#### 3.5.3. Developing Standards of Quality and Accessibility

Interviewees also mentioned the need for standards.

*Assistive technology and devices are very important for persons with disability. But what we see when we go to the government departments like the social welfare department, the health department, or charitable organizations that donate assistive devices—What I see in India is that it is not sustainable, while these products may be distributed, they are of very poor quality so they last for a very short time. Sometimes I even call them disposable items. People can use these assistive technologies and devices for three to six months and after that again have to suffer for a long time without them. Government of India must ensure that there are better standards for assistive technology and devices available for persons with disabilities so that we can use these products at least for five to ten years we don’t need replacement* (I4, India).

One respondent made this point when it came to communication,

*The biggest challenge is that our government has still not prioritized the provision of CAPTIONS for broadcast services, this, in spite of the mandate being given to ICASA to implement the provision of CAPTIONS by 2019. ICASA as the implementing agency has missed its deadline and so this has resulted in people with hearing loss not having access to information* (I7, South Africa).

#### 3.5.4. Lack of Government Understanding

Additionally, in an overarching theme, several interviewees spoke about the fact that the government did not understand their concerns or had a limited view of the different types of difficulties AT users experience and their specific needs as a result. They emphasized that their voices were needed in fashioning these policy reforms.

*The government doesn’t have an understanding about the needs of blind people. They think disability is wheelchair persons or just providing Braille* (I21, South Africa).

*I don’t know the health coordinator in my community responsible for me. Right now, it’s only the Chennai cooperation zonal office and they don’t know me and my requirements or the requirements of other persons with disabilities or older persons in the community (…) Therefore, I suggest there should be sensitized community support system and well-informed staff and general public* (I10, India).

Interviewees felt that people with specific needs such as theirs should be involved in the planning, implementation, and monitoring of all government responses to a pandemic—or to any policy concern. Their specific knowledge of what it means to live in their society is an invaluable component of policy development [14], both from a practical standpoint but also from a moral one to the extent, as expressed in the Convention on the Rights of Persons with Disabilities, and other human rights instruments that all people should have a say in how society is structured.

Notably, this point and the policy suggestions by interviewees listed below have been echoed in other recent literature by policy analysts [15,16]. This is true in particular about the lack of planning.

The results of 35 interviews over six global regions described above presents a detailed view of the intersecting factors (both barriers and enablers) affecting life during COVID-19 for AT users. This rich picture, and the thematic imperatives arising from the data, is further discussed in light of the literature, to frame our concluding recommendations.

## 4. Discussion

Current research on COVID-19 is telling us that the pandemic is having a disparate impact on people who use AT, not only because of their functional limitations but the impact of its isolating effects [17]. Another finding from this literature is the importance of AT in ameliorating that impact [18]. However, current AT policies and systems have by and large not met the challenge [4,19]. While some evidence suggests that AT provision can be supported through family and social networks, risks remain for people who use AT with the switch to telehealth. Our research drawing on in-depth interviews with a range of AT users from six middle- and low- income countries affirms these results and provides added depth to the understanding of these concepts.

In general, our results show the importance of an inclusive approach at all points in the AT system. People with disabilities must be included throughout the system. This includes clear lines of accessible communication between users and providers. Supply chains and product quality must be made resilient to crises. All of this requires strong policies and coordination with civil society responses.

In presenting these results, we follow the WHO GATE 5P Framework, which provides a model for examining AT through five key intersecting dimensions. These include *people* (AT users), *provision* systems, and *personnel* (providers) who supply assistive *products* within relevant *policy* landscapes [20,21]. This discussion is organized around those dimensions.

### 4.1. People

AT users, including people with disabilities, health conditions, and older persons, have particular needs and a particular perspective on service delivery [22,23,24,25]. These necessitate an inclusive response in crises, such as the COVID-19 pandemic. Their representation in developing, evaluating, and planning for response is vital for making sure that such events do not cause significant disruption in their lives and the lives of their families. The Convention on the Rights of Persons with Disabilities specifically references people with disability, who are often users of AT, and this representation is critical.

### 4.2. Personnel

Access to, and capabilities of, AT-related personnel are well documented [24] In pandemic conditions, a range of personnel-related issues became apparent [25,26,27,28]. Timely, reliable information is essential to respond to a crisis such as COVID-19 in the best manner possible, both for governments and for individuals. Governments and media outlets must have the personnel capable of providing this. At the outset of the pandemic, information was needed on the extent and nature of the health crisis, but also on what steps the public should take to protect themselves. This involved information on how the virus was transmitted, how people could prevent infection by sanitizing their hands and surfaces, and how to use masks. Furthermore, people who use AT often face barriers to communication, both because of accessibility issues and the fact that being, on average, poorer they may have less access to the internet [29,30]. Interviewees reported government attempts to provide them with the information they needed in accessible formats were often problematic. To ensure accessible information, government communication should be accessible to all. This includes the provision of phones and communication apps. Accessible phones should be considered as an assistive product for those with communication difficulties.

### 4.3. Products

Poor-quality AT that may easily break in the middle of a crisis when it is hard to repair or replace can lead to injury to the person using it, or a loss of function if the assistive product is no longer usable. During crises when maintenance and repair are more difficult, the need for strong-quality standards is even greater. The data resonated with the position on assistive products from the first global research, innovation, and education on assistive technology (GREAT) summit, Disability and Rehabilitation: Assistive Technology [31], and a crisis management lens applied to this position paper may address the type of shortfalls in product supply and maintenance noted in this study. This is especially important in places that do not have local production and rely on external supply chains. Ideally, governments should establish quality and accessibility standards for all types of goods and services and ensure that what they provide is usable and effective.

### 4.4. Provision

An important part of planning for future global health crises is developing and implementing standards of accessibility for AT services to ensure that when the time comes to act that those actions can actually have the desired impact. Data demonstrated that AT, for example, must be available, affordable, good quality, properly fitted, and properly maintained, and this is echoed in the literature [12]. Provision without these elements can lead to wasted time and resources without delivering the desired impact. Services being more widely distributed and closer to people’s homes would have assisted during lock downs, disrupted travel, and difficulties accessing public transport.

### 4.5. Policy

The provision of AT can be complex and varied. AT is often vital to people’s lives and impacts not only them, but their whole family. The financial cost of accessing AT can be significant, but the costs to people needing it and not having access can be severe, including economic and social dislocation. A patchwork approach to AT provision will not be sufficient. Governments can play a vital role by having well-thought out, coordinated policies that assure good information, access to AT, and by providing cash and in-kind benefits to people suffering hardship from the pandemic [18].

AT needs to be recognized as an essential component of health service delivery, and stronger messaging and prioritizing of these services is required in situations where health services are being limited in response to a pandemic.

Inclusive disaster risk management is important [32]. Proper preparation includes population-based data that highlight areas where people with disabilities may be disproportionately located and, thus, where AT needs and related support services may be higher during a crisis. This preparation needs to include people with disabilities [33].

The results from these interviews were in line with the results found from the surveys collected simultaneously from a broader sample [13]. The recommendations from that arm of this study, as well as the data from AT providers [2], made clear the need for better planning and implementation involving the consultation of AT users, the need for pandemic responses to recognize the importance of AT, and to strengthen the delivery systems of AT. The interviews reported on here confirm those findings. They also more clearly show the need for accessible information, more inclusive pandemic (and other public health and disaster) preparedness planning, and the needs for AT standards of quality and accessibility. They demonstrate the civil society response that can occur when government response is insufficient. Table 3 contains the full set of recommendations drawn from across the three studies.

While the results led to clear recommendations, it must be kept in mind that as qualitative analysis, the prevalence and extent of the concerns raised in the interviews cannot be reported at a population level. The methodology does not lead to an estimate of how often these problems occurred in the population or how they are correlated with characteristics at the population level. However, they are strong indications of a range of people’s experiences and the common themes that emerged in the interviews.

## 5. Conclusions and Future Perspectives

The conclusion drawn from a series of interviews of people who use AT in low- and middle-income countries is that the government response to the pandemic, both in providing information and in delivering services, was insufficient and not well adapted to their needs. In many instances, non-government organizations filled this gap. While their efforts were appreciated, they were of essence more concentrated in urban areas and among people who were well connected to disability and/or other community level networks. A government response that can be nationwide and prepared in advance would no doubt be more effective. However, that preparation needs to be undertaken with direct representation of people with particular needs including those who use AT to ensure that it meets their needs.

## Figures and Tables

**Table 1 ijerph-18-12031-t001:** Overview of themes and sub-themes identified.

Theme	Sub-Theme
Information	Civil society
	Family and friends
	Social media
	Phones
	Government
	Television and radio
AT	AT: Cost, customization/repair
	Support: Personal assistants, mental health, transportation
Personal Resilience	Education Family and friends Meditation
Post-COVID-19 Impact	Economic: access to jobs, affording services, workplace adaptations
	Health Services: access to services, new health problems
	Social: Discrimination, isolation, mental health
Supports	Lack: Family, government, hospitals, private sector Provided: Civil society, family, government
Policy Reforms	Communication Community level Policy planning Standards

**Table 2 ijerph-18-12031-t002:** Respondent ages and types of functional difficulties.

Characteristic	Type	Number
Age	0–18	2
	19–29	6
	30–39	10
	40–49	9
	50–59	4
	60+	4
Type of functioning difficulty	Communication	1
	Cognition	1
	Hearing	7
	Mobility	16
	Self-care	1
	Vision	11
	Psychosocial	4

**Table 3 ijerph-18-12031-t003:** Recommendations.

Recommendations
**1.** **Make pandemic public health responses inclusive of people who use AT:** Consult with civil society including AT users, their families, and representative bodies Understand and mitigate against the impact health responses may have on people who use AT Use communication formats that ensure public health messages are accessible to all, including people with hearing or vision loss, or disabilities that impact cognition and/or communication Recognize information and communication technologies, including smart phones, as priority assistive products. **2.** **Recognize AT as essential health products and services and during a pandemic or health emergency:** Keep AT services open, safe, and accessible alongside other essential services Consult with and include AT personnel in health sector wide responses Provide AT personnel with infection control training and personal protective equipment Implement telehealth and other methods that enable services to continue during pandemic response measures such as isolation, social distancing, and/or lockdown Prioritize continued procurement and supply of quality-assured assistive products **3.** **Strengthen AT services to improve preparedness for future pandemic responses:** Integrate AT services into health care systems and, in particular, community/primary health care Address access barriers and increase coverage through outreach visits, telehealth, and other strategies Train and equip a broader range of health personnel able to provide and/or support AT use

## Data Availability

Not applicable.

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
