# Peer review of "Assistive Technology during the COVID-19 Global Pandemic: The Roles of Government and Civil Society in Fulfilling the Social Contract"

_ijerph, 2021, doi:10.3390/ijerph182212031_

Round 1
Reviewer 1 Report
The paper addressed a very central topic and it is well designed and well described. I suggest just some minor changes.
First, to better discuss results according to different kind of impairments declared (if possible).
Second, to discuss differences according to the country of each individual considered, also with reference to different approach/approaches used by each country in coping with the COVID19 outbreak.
Third, to discuss practical consequences of findings.
Author Response
Thank you for your comments.
Reviewer 2 Report
I strongly suggest evaluating the CROSS guidelines https://doi.org/10.1007/s11606-021-06737-1
Introduction
28-36 references missing. [ https://www.mdpi.com/1660-4601/18/19/10477 ; https://dx.doi.org/10.1016%2FS0140-6736(21)00625-5 ; https://dx.doi.org/10.1108/JET-11-2020-0047 ; https://dx.doi.org/10.34172/ijhpm.2020.210]
I believe it is necessary to describe the AT before outlining the needs and limits.
49-68 I recommend arriving at the last sentence of the introduction with a solid and concise objective of 10 lines maximum
Methods
Specify the study design in the “Methods” section with a commonly used term (e.g., cross-sectional or longitudinal).
Describe the questionnaire (e.g., number of sections, number of questions, number and names of instruments used). Moreover, I suggest eliminating that draft of the literature review, I think it is satisfactory to describe the genesis of the survey, if a board of experts validated it or if the questionnaire refers to other references in the literature.
I would suggest for Table 1, to insert the actual questionnaire instead of the "Overview of themes and sub-themes identified"
Report target population, reported validity and reliability information, scoring/classification procedure, and reference links (if any). So I suggest drafting some sort of eligibility of the population to which the questionnaire was administered. Describe the study population (i.e., background, locations, eligibility criteria for participant inclusion in survey, exclusion criteria). Moreover, Describe the sampling techniques used (e.g., single stage or multistage sampling, simple random sampling, stratified sampling, cluster sampling, convenience sampling). Specify the locations of sample participants whenever clustered sampling was applied.
Most of all: Describe statistical methods and analytical approach. Report the statistical software that was used for data analysis
Results
Report numbers of individuals at each stage of the study. Consider using a flow diagram, if possible.
I would suggest tabulating the responses from the various countries. Table 2, what is meant by functional difficulty? A reference to the ICF should be described in the methods and it would be better to outline the eligibility.
Results are long verbose; I recommend a severe volume reduction. Above all, I recommend tabulating the results by giving "an unadjusted estimates and, if applicable, confounder-adjusted estimates along with 95% confidence intervals and p values.”. Quite simply tabulate the questions already described in the methods with a lean table Q1, Q2, Q3 ...
I recommend once again to be strict in describing the results. In the discussion each question can be argued with greater emphasis.
Discussion
I recommend starting after the goal with the major findings of the study. The limitations section is completely missing. I suggest removing the recommendations and moving them to the concluding part in order to describe them based on the limitations of the investigation.
Conclusion
Given the recommendations, I would suggest redefining the section as Conclusions and future perspectives
Author Response
Thank you for your comments.

Round 2
Reviewer 2 Report
I congratulate you on your efforts in revising, I feel I can affirm that the manuscript is suitable for publicationI apologize for the quantitative approach given to the method, if I can make a final note I would add some bibliographic references in more discussion:
People: https://doi.org/10.1080/13691457.2021.1882397 https://doi.org/10.1016/j.maturitas.2020.04.004
Personnel https://doi.org/10.1017/S0033291720001622 https://doi.org/10.3390/ijerph18189676 https://doi.org/10.1007/s00406-020-01183-2
Policy https://doi.org/10.1016/S0140-6736(21)01233-2 https://doi.org/10.1007/978-981-15-7679-9_22
Author Response
Response to Reviewer #2 on Second Revision
Thank you for reviewing again, and for the additional citations which add to placing our paper in the context of the current literature. We have included them where they were most appropriate and included an additional sentence at line 666 to add context to one of them, shown below.
This preparation needs to include people with disabilities [38].
